# Tool Run-Out in Micro-Milling: Development of an Analytical Model Based on Cutting Force Signal Analysis

**DOI:** 10.3390/mi15030305

**Published:** 2024-02-23

**Authors:** Andrea Abeni, Cristian Cappellini, Greta Seneci, Antonio Del Prete, Aldo Attanasio

**Affiliations:** 1Dipartimento di Ingegneria Meccanica ed Industriale, Università degli Studi di Brescia, Via Branze, 38, 25123 Brescia (BS), Italy; andrea.abeni@unibs.it (A.A.); greta.seneci@unibs.it (G.S.); 2Dipartimento di Ingegneria Gestionale, dell’Informazione e della Produzione, Università degli Studi di Bergamo, Via Pasubio, 7/b, 24044 Dalmine (BG), Italy; cristian.cappellini@unibg.it; 3Dipartimento di Ingegneria dell’Innovazione, Università del Salento, Complesso Ecotekne–edificio “Corpo O”, Via per Monteroni, 73100 Lecce (LE), Italy; antonio.delprete@unisalento.it

**Keywords:** tool run-out, micro-machining, cutting force model, LB-PBF

## Abstract

Micro-machining is a widespread finishing process for fabricating accurate parts as biomedical devices. The continuous effort in reducing the gap between the micro- and macro-domains is connected to the transition from conventional to micro-scale machining. This process generates several undesired issues, which complicate the process’s optimization, and tool run-out is one of the most difficult phenomena to experimentally investigate. This work focuses on its analytical description; in particular, a new method to calibrate the model parameters based on cutting force signal elaboration is described. Today, run-out prevision requires time-consuming geometrical measurements, and the main aim of our innovative model is to make the analysis completely free from dimensional measurements. The procedure was tested on data extrapolated from the micro-machining of additively manufactured AlSi10Mg specimens. The strategy appears promising because it is built on a strong mathematical basis, and it may be developed in further studies.

## 1. Introduction

One of the most thriving areas of industrial research in recent times is the additive manufacturing (AM) of metallic alloys. The AM of metals includes a large collection of different processes, which can be classified by considering how the raw material is supplied (powder or wire or sheet shape) or how the energy is provided (laser, electron beam, or arc) [1]. AM finds several applications in different industries, but biomechanics and aerospace are the sectors where AM is most employed. Light-weight and biocompatible aluminum alloys stand out among processable alloys, and they are also the second most widely used metallic structural material after steel. Aluminum alloys have broad potential for applications in aerospace, automotive, defense, electronics, and biomedical industries, but the performances of parts produced by the traditional casting process are poor [2]. Traditional casting and its low cooling rates cause macro-segregation and coarse structures in aluminum alloys [3]. To overcome this issue, together with the need for a long production chain and the limited flexibility of plastic processing, numerous studies on AM can be found in the literature. In [4], an in-depth analysis of stereolithography (STL) on AlSi10Mg was proposed, while in [5], the same topic was analyzed with a micro-image-based (by X-ray tomography) finite-element simulation, which led to a model for quantifying process-induced defects. The laser-based powder bed fusion (LB-PBF) of AlSi10Mg has been successfully commercialized and has become the most popular additive manufacturing technology for Al alloys owing to its low density and high specific strength [6]. 

Once an AM complex functional part is produced, it needs further finishing to make the component ready for critical application (aerospace, automotive, biomedical, optical, military, etc.). This is an example of how different technologies are now combined to exploit their advantages to achieve new properties or uses.

Another important aspect is miniaturization and the high precision requirements of manufacturing processes [7]. Among all of the precision production technologies, micro-milling is one of the most studied because it is promising for accurate feature fabrication. To turn this technology into a large-scale production process, its predictability requires further development. Micro-milling is defined as the result of the scaling down from conventional sizes (feed rates of hundreds of micrometers per tooth, depths of cut equal to several millimeters) to micro-end-milling sizes (feed rates of several micrometers per tooth, depth of cut of hundreds of micrometers) [8], and its accuracy is lower than 1 µm. The integration between AM and micro-machining leads to the definition of hybrid additive manufacturing, which refers to material removal being performed after the 3D fabrication of near-net-shaped products [9]. Micro-machinability at a low dimensional scale is strongly affected by the microstructure and the derived mechanical properties [10] due to the size effect; additively manufactured alloys exhibit specific properties due to the different chemical composition and process conditions, such as localized elevated thermal loads and the consequent high-rate cooling. There is wide literature on the machinability of LB-PBF samples. Owing to the rapid development in hybrid additive manufacturing, reference [11] proposed a comparison between the machinability of cast and LB-PBF samples produced in AlSi10Mg and subjected to T6 solution heat treatment. In [12], another alloy for orthopedic application was investigated to build a numerical model using the DEFORM-3D v11.3 commercial software. The model integrated the experimental cutting parameters and it was used for predicting the formations of serrated chips, chip thickness, and tool wear mechanism. Several studies have been conducted on the final quality of micro-machined parts; in [13,14], surface roughness was investigated and the correlation with the process parameters was discussed. 

While traditional milling and micro-milling are equivalent from a kinematic point of view, the main issues connected to the change in sizes are the undesirable phenomena observed at the micro-scale. In the micro-milling of the tool edge radius dimension, the uncut chip thickness and the alloy grain size are of the same order of magnitude, so there is a need to define a new model of the process. In [15], a thorough review about all micro-milling issues, including micro-burr formation mechanisms, cutting temperature, vibrations, surface roughness, cutting fluids, and cutting forces, is available. In particular, the ratio of feed per tooth to the radius of the cutting edge is lower in micro-milling than in conventional machining. The low ratio promotes a negative rake angle, which can cause an incorrect material removal mechanism known as the ploughing condition. Ploughing has negative effects on surface integrity; furthermore, it increases the difficulties in predicting cutting forces. In [15], one of the first numerical attempts to model the effects of temperatures and chip formation on cutting forces during micro-machining operations was performed. In [16], another mechanistic cutting force model was presented, and it considered the effect of high rotational speeds, which amplify tool run-out.

In the cutting force prediction for micro-machining processes, the tool run-out phenomenon plays a primary role. Tool run-out determines the difference between the tool edge’s actual and theoretical trajectories [17]. It greatly affects the accuracy of micro-milling, in contrast to conventional milling, where the same phenomenon has neglectable effects. Tool run-out leads to increased tool wear [18] and a reduction in the surface quality for the finished parts [19], and it makes force prediction more difficult. An analytical model for predicting the surface topology when micro-milling three different materials, namely 40HM steel, Al7035 aluminum alloy, and Ti-6Al-4V titanium alloy, was proposed in [20]. The authors revealed that roughness was mainly influenced by the cut width, while a significant effect of the feed was observed only at low cut width values. In [21], a wide description of this topic is presented with a geometrical description; the parameters were the same as those used in this article. The important result in [21] is the experimental method proposed to measure tool run-out from the cutting force signal. Once the mill diameter, the cutting edge phase angle α, and the rotational radius of the first cutting edge are known, the tool run-out parameters could be estimated. It used the cutting period derived from the cutting force signal to determine α, while for determining the first cutting edge radius the physical measurement of the width of the cut is mandatory. A cutting energy-based model capable of predicting run-out geometry is developed in [22], where good forecasting of machined surface roughness is achieved. In [23], an experimental technique, exploiting the usage of a boring tool holder for evaluating the tool offset modification, is presented. The implementation of a piezo actuator for compensating tool run-out using workpiece displacement improved the accuracy of the machining operation. As all the experimental procedures required to describe the run-out effect are time-consuming, in [24] a valid analytical model to estimate cutting forces is presented; it is applied to a wide range of process parameters, specifically for Inconel samples produced from the AM process. A more complex force predictive model is presented in [25]. It deals with another significant phenomenon in micro-milling: the transition between the ploughing regime and the shearing one. The shearing occurs only when the actual chip thickness increases, and it reaches the minimum uncut chip thickness (*MUCT*). In [26], an analytical model for the evaluation of the asymmetrical behavior of machining forces in micro-milling operations is presented and it considers both the transition between ploughing and shearing regime in cutting and the effects of tool run-out. In [27], the first analytical model for cutting force prediction deals with the transition between the regimes. A tool run-out model based on the replacement of location and tilt tool angle with the tool axis direction vectors, allowing good prediction of instantaneous uncut chip thickness (*IUCT*) and forces, is presented in [28]. In other research [29], a dynamic force model based on the instantaneous stiffness is introduced, permitting the estimation of run-out by calibrating it with the modeled forces. In [30], the aim is not the force prevision, but there is a useful uncut chip thickness model that considers the precise trochoidal trajectory of the cutting edge, tool run-out, and dynamic modulation caused by the machine tool system vibration. Always based on *IUCT*, the effect of tool run-out on the behavior of cutting forces was studied in [31] by flank shearing, ploughing, and bottom edge coefficient calibration. The instantaneous calibration of cutting force coefficients in the presence of tool run-out is also discussed in [32], giving encouraging results.

All the studies previously mentioned are valid for the prediction of cutting forces in micro-milling and they consider the effects of the specific phenomena related to the micro-scale. Nevertheless, the data collection of those methods requires expensive online experimental acquisition devices correlated to direct physical time-consuming offline measurements. Currently, in the literature, there is a lack of models that allow the prevision of run-out from an analytical point of view and without direct measurements on the machined parts. In this work, the force signal directly acquired from micro-milling experiments is used to predict run-out, permitting a future implementation for tool path monitoring and compensation in an online mode. The model reliability is tested by comparing the model results to the experimental ones. It is also mathematically demonstrated that the proposed model applies to different materials. The demonstration was performed with a sensitivity analysis which confirmed the independence of the model from the value of the force-specific constant K_ts_. 

## 2. Materials and Methods

Tool run-out is a not neglectable phenomenon in the study of the micro-machining process due to the extreme accuracy usually required. For this reason, the modeling of micro-machining must consider the geometrical description of the tool run-out. It allows us to derive some relations between the run-out parameters and to compute a mathematical representation of the process. The model described in this work makes the tool run-out prediction independent from time-consuming experimental measurements: the only variables that it requires are the process geometrical features and the force signals [33,34] automatically recorded during the milling process. The aim of the second part of this section is the validation of the model so that the extensive experimental campaign is described, and it is used to compare the run-out predicted by the elaboration from the force signal with the actual run-out computed by measuring the machined features.

### 2.1. Proposed Model

#### 2.1.1. Tool Run-Out

Tool run-out is the result of the sum of many different phenomena, among which are the geometrical displacements of the spindle axis, tool-holder axis, and tool axis from the theoretical rotation axis. 

Figure 1 represents the run-out effect on the tool together with the geometric main parameters which must be considered in the discussion [21]. The two tool-cutting edges are sectioned in a plane normal to the theoretical tool rotation axis and parallel to the feed direction. 

The two main run-out parameters are: *r*_0_, the distance between the spindle’s rotational center O and the tool one O’;*γ*_0_, the angle from *r_ce_*_1_ and *r*_0_.

They cannot be experimentally measured, but there are mathematical equations to compute them from other geometrical variables (Equations (1)–(4)). Those equations are widely discussed in Attanasio’s work [21]:
(1)
$$r_{CE1}=\underline{OA}=(\frac{d}{2}+r_{0})\cdot \sqrt{1+\frac{d\cdot r_{0}\cdot (cos\gamma -1)}{{\left(\frac{d}{2}+r_{0}\right)}^{2}}}$$


(2)
$$r_{CE2}=\sqrt{r_{CE1}^{2}+d^{2}-2r_{CE1}\cdot d\beta \;}$$


(3)
$$\delta =arcsin(\frac{\underline{OA}}{\underline{AB}}\cdot sin\;\alpha \;)$$


(4)
β=π−α−δ


The variables included in the previous equations are:the tool diameter *d* [mm];the rotational speed *ω* [rad/s];the cutting edge phase angle *α* [rad].

The only term that is required to be experimentally found for its use in the final equation is *r_CE_*_1_ [mm], the rotational radius of the main cutting edge. The tool run-out causes the variation of two parameters previously described:angle *α* would not be constantly equal to *π* [rad];radius *r_CE_*_1_ would differ from the rotational radius of the second cutting edge *r_CE_*_2_ [mm].

#### 2.1.2. Analytical Model

In [21], further equations are presented to compute all the geometrical quantities visible in Figure 1. In [21], there are also the equations to describe the cutting trajectories of the two tool edges (Equations (5)–(9)). The term called *θ*(*t*) is the tool rotational angle which is a function of time (Equation (5)). In the following equation, *CE*1 stands for the first cutting edge while the subscript *CE*2 is used to refer to the second cutting edge.

(5)
$$\theta \left(t\right)=\omega \cdot t$$


(6)
$$x_{CE1}=r_{CE1}\cdot sin\;\theta +\frac{f}{60}\cdot t$$


(7)
$$y_{CE1}=r_{CE1}\cdot cos\;\theta$$


(8)
$$x_{CE2}=r_{CE2}\cdot sin\;\left[\theta +\alpha \right]+\frac{f}{60}\cdot t$$


(9)
$$y_{CE2}=r_{CE2}\cdot cos\;\left[\theta +\alpha \right]$$


In [26], the previous equations are derived to estimate the instantaneous uncut chip thickness (*IUCT*) (Equations (10)–(13)). Each one of the two mill cutting edges has its value for *IUCT* and they are called *h_CE_*_1_ and *h_CE_*_2_ (Figure 2a). In the following functions, Δ*s_CE_*_1_ is the distance crossed by the rotational axis between the passage of the second cutting edge *CE*2 and the consecutive passage of the first one, *CE*1, in the *θ* instantaneous angular position; vice versa, the value Δ*s_CE_*_2_ is the distance defined from the passage of the rotational axis of *CE*1 and the consecutive passage of the one for *CE*2_,_ always in the *θ* angular position.

(10)
$$h_{CE1}=\sqrt{{\left(r_{CE1}sin\;\theta +{∆s}_{CE1}\right)}^{2}+{\left(r_{CE1}cos\;\theta \;\right)}^{2}}-r_{CE2}$$


(11)
$$h_{CE2}=\sqrt{{\left(r_{CE2}sin\;\theta +{∆s}_{CE2}\right)}^{2}+{\left(r_{CE2}cos\;\theta \;\right)}^{2}}-r_{CE1}$$


(12)
$${∆s}_{CE1}=\frac{f}{60}\cdot \frac{\alpha }{\omega }$$


(13)
$${∆s}_{CE2}=\frac{f}{60}\cdot \frac{2\pi -\alpha }{\omega }$$


Then, the cutting area of the i-th cutting edge is determined (Equation (14)) to be used later in the calculation of ploughing areas; indeed, two of the variables in the cutting force equation are the ploughing areas and not the overall cutting area.

(14)
$$A_{ci}\left(\theta \right)=\int_{0}^{\theta }\left(\frac{h_{CEi}\left(\theta \right)+h_{CEi}\left(\theta +d\theta \right)}{2}\right)\cdot r_{CEi}\cdot d\theta$$


While the cutting area *A_ci_* is the total surface of material removed by a single cutting tool edge, *Ap_CE_*_1_ and *Ap_CE_*_2_ are defined as the ploughed area for the two cutting edges [mm^2^] (see Figure 2a). The ploughed area is the portion of the cutting area where the material is deformed and not cut. The so-called minimum uncut chip thickness (*MUCT*) must be used to detect the transition from the ploughing regime to the shearing one. As Equations (15)–(17) show [25], when *MUCT* is higher than *h_CEi_*(*θ*) the cutting area of the i-th edge is equal to the ploughing one. From Equation (16), it is possible to state that when *MUCT* is lower than the *h_CEi_*(*θ*) value, the i-th cutting edge’s ploughing area (*A_pCEi_*) is equal to the maximum one among all the cutting edges (*A_pMAX_*). Another result is that if the value of *MUCT* is lower than *h_CEi_*(90°), the cutting area for the edge is the maximum possible and it is equal to the ploughing area.

(15)
$$h_{CEi}\left(\theta \right)<MUCT\;⋀\;\theta <\theta_{MAXi}\;\to \;A_{pCEi}\left(\theta \right)=A_{ci}\left(\theta \right)$$


(16)
$$hCEi\left(\theta \right)>MUCT\;\to \;A_{pi}\left(\theta \right)=A_{piMAX}$$


(17)
$$h_{CEi}\left(\theta \right)<MUCT\;⋀\;\theta >\theta_{MAXo}\;\to \;A_{pCEi}\left(\theta \right)=A_{ci}\left(\pi \right)-A_{ci}\left(\theta \right)$$


The total cutting force *F_c_* is the combination of its axial components *F_x_*, *F_y_*, and *F_z_* in the three dimensions. In Figure 2a, the *F_z_* component is not visible because it is orthogonal to the section plane. The resulting *F_c_* could also be subdivided into its tangential *F_t_* and radial *F_r_* components; the tangential direction follows the cutting edge *θ* generic angular position and it is defined in a plane orthogonal to the tool axis. These last force components can be expressed by Equations (18)–(21) for the two cutting edges:
(18)
$$F_{t1}=\left(K_{ts}\cdot h_{CE1}\left(\theta \right)+K_{tp}\cdot A_{pCE1}\left(\theta \right)\right)\cdot a_{p}$$


(19)
$$F_{r1}=\left(K_{rs}\cdot h_{CE1}\left(\theta \right)+K_{rp}\cdot A_{pCE1}\left(\theta \right)\right)\cdot a_{p}$$


(20)
$$F_{t2}=\left(K_{ts}\cdot h_{CE2}\left(\theta \right)+K_{tp}\cdot A_{pCE2}\left(\theta \right)\right)\cdot a_{p}$$


(21)
$$F_{r2}=\left(K_{rs}\cdot h_{CE2}\left(\theta \right)+K_{rp}\cdot A_{pCE2}\left(\theta \right)\right)\cdot a_{p}$$

where:*a_p_* is the axial depth of cut [mm];*K_ts_* and *K_rs_* are the specific force coefficients for the shearing regime [N/mm^2^];*K_tp_* and *K_rp_* are the specific force coefficients for the ploughing regime [N/mm^3^].

Once the force tangential and radial components are determined, in Equations (22)–(25) there is the mathematical passage to the force components for the *x*-axis and *y*-axis.

(22)
$$F_{x1}=F_{t1}\cdot cos\;\theta +F_{r1}\cdot sin\;\theta \;$$


(23)
$$F_{y1}={-F}_{t1}\cdot sin\;\theta +F_{r1}\cdot cos\;\theta \;$$


(24)
$$F_{x2}=F_{t2}\cdot cos\;\theta +F_{r2}\cdot sin\;\theta \;$$


(25)
$$F_{y2}={-F}_{t2}\cdot sin\;\theta +F_{r2}\cdot cos\;\theta \;$$


Considering that the peak force of the two cutting edges will be in correspondence with maximum chip cross-sectional area, for *CE*1 the peak will be when *θ* ≅ 90° while for CE2 the peak will be when *θ* ≅ 270° (Figure 2b). Introducing those values in Equations (22)–(25) and using the definitions for the force components in Equations (18)–(21), a new function (Equation (26)) is found to define the peak difference for the y component (it is Δ*F_y_* in Figure 2b); the y-axis is orthogonal to the feed direction.

(26)
$$\frac{∆Fy\left(90{}^{\circ}\right)}{a_{p}}=K_{ts}\left(h_{CE1}\left(90{}^{\circ}\right)-h_{CE2}(90{}^{\circ})\right)+K_{tp}\left(A_{pCE1}\left(90{}^{\circ}\right)-A_{pCE2}\left(90{}^{\circ}\right)\right)$$


The following hypothesis is assumed: the ploughing areas for the two cutting edges could be considered equal. This statement is consequential of the use of a high-precision tool holder which will determine a small value of run-out; a small run-out is the result of cutting edge trajectories being very similar (*r_CE_*_1_ equal to *r_CE_*_2_ and Δ*s_CE_*_1_ equal to Δ*s_CE_*_2_), so the ploughing areas will be the same. Equations (15)–(17) exploit that if *h_CE_*_1_(90°) is higher than *MUCT*, the ploughing area of the first cutting edge is equal to the maximum ploughing area among all the tool cutting edges; in the study case, the cutting edges are only two, so *A_pCE_*_1_(*θ*) is equal to *A_pCE_*_2_(*θ*). The hypothesis that *A_pCE_*_1_(*θ*) is equal to *A_pCE_*_2_(*θ*) allows us to simplify Equation (26) by writing Equation (27):
(27)
$$\frac{∆Fy\left(90{}^{\circ}\right)}{a_{p}}=K_{ts}\left(h_{CE1}\left(90{}^{\circ}\right)-h_{CE2}(90{}^{\circ})\right)$$


In the passage from Equation (26) to Equation (27), all the specific force coefficients previously presented can be neglected except for *K_ts_*; its definition needs a sensitivity analysis to investigate its effect on the final analytical equation. 

As the *θ* value is fixed, the variable Δ*F_y_* becomes independent from the angular position and an angle of 90° leads to the maximum of the y component peak difference (Δ*F_ymax_*). At the same time, when *θ* reaches the value of 90°, the functions to find *h_CE_*_1_ and *h_CE_*_2_ can be rewritten and Equation (28) is determined.

(28)
$$\frac{∆Fymax}{K_{ts}{\cdot a}_{p}}=2\cdot \left(r_{CE1}-r_{CE2}\right)+{∆s}_{CE1}-{∆s}_{CE2}$$


By applying the definitions of Δ*s_CE_*_1_ and Δ*s_CE_*_2_ presented in Equations (12) and (13), Equation (28) can be simplified in Equation (29):
(29)
$$r_{CE1}-r_{CE2}=I=\frac{∆Fymax}{{2\cdot K}_{ts}{\cdot a}_{p}}-\frac{f}{60\cdot \omega }\left(\alpha -\pi \right)$$


The difference between the two cutting radii will be called in all the further equations variable *I*. Variable Δ*F_yMAX_* can be estimated by the analysis of the cutting force signal. Substituting in Equation (29) the values of *r_CE_*_2_ and *β*, the final 4th grade equation to estimate *r_CE_*_1_ is found (Equation (30)). All the variables written in the model are described in Equations (31)–(35).

(30)
$$a\cdot {r_{CE1}}^{4}+b\cdot {r_{CE1}}^{3}+c\cdot {r_{CE1}}^{2}+e\cdot r_{CE1}+g=0$$


(31)
$$a={sin}^{4}\left(\alpha \right)+{cos}^{2}\left(\alpha \right)\cdot {sin}^{2}\left(\alpha \right)$$


(32)
$$b=-2\cdot I\cdot {sin}^{2}\left(\alpha \right)$$


(33)
$$c=I^{2}+I^{2}\cdot {sin}^{2}\left(\alpha \right)-d^{2}\cdot {sin}^{2}\left(\alpha \right)-d^{2}\cdot {cos}^{2}\left(\alpha \right)$$


(34)
$$e=I\cdot d^{2}-I^{3}$$


(35)
$$g={\left(\frac{I^{2}}{2}-\frac{d^{2}}{2}\right)}^{2}$$


Equation (30) allows us to compute *r*_*CE*1_ and, once it is known, the geometrical model of tool run-out is defined. The previous approach described in [26] required the experimental determination of *r_CE_*_1_ by measuring the width of the microchannels and by assuming that the width is equal to double *r_CE_*_1._ In this work, both methodologies were applied to compare the results to validate the innovative approach. In the case that the experimental results would be coherent with the values predicted by Equation (30), the hypothesis on which the innovative approach is based can be considered valid. 

### 2.2. Experimental Campaign

The experimental campaign consists of micro-milling of slots in AlSi10Mg alloy specimens fabricated via LB-PBF. In Table 1, there is the chemical composition of the alloy. 

#### 2.2.1. Specimen Preparation

The prismatic specimen used in the tests was built through laser-based powder bed fusion (LB-PBF). The additive machine used for the specimen production is an EOS M290 (EOS GmbH, Krailing, Germany) and in Table 2 the process parameters are listed. The specimen was also subjected to two further treatments: hot isostatic pressing (HIP) followed by T6 heat treatment. The HIP mechano-thermal treatment consists of heating to 520 °C for 2 h while the T6 heat treatment consists of a solution treatment at 540 °C for 7 h, followed by quenching in water and artificial aging at 160 °C for 4 h.

The sample shape was designed to constrain the sample itself on the load cell; so, two small holes have been made at the specimen edges (see Figure 3). The surface roughness was measured by using a laser profilometer (Mitaka PF60, Mitaka Kohki Co., Ltd., Tokyo, Japan) by repeating three measurements. It resulted in Ra = 16.9 ± 4.1 µm and Rz = 89.9 ± 21.4 µm.

#### 2.2.2. Micro-Milling Experiments

The tool used in the micro-milling experiments is a Ø0.8 mm flat-bottom two-flute micro-mill. In Table 3, there are the main features of the tool. The tool diameter was measured three times before every milling test and by using a multifocal microscope (Hirox RH-2000, Hirox Japan Co., Ltd., Tokyo, Japan). 

In Figure 3, there is a representation of the micro-slot scheme. As Figure 3b shows, 15 micro-slots have been fabricated during the milling experiments on the AM specimen. In the pattern of Figure 3b, it is possible to see 20 slots because excess space was designed to have the possibility to repeat tests in cases of issues about the cutting force data acquirement. The cutting parameters used in the milling tests are listed in Table 4. Fourteen values of feed per tooth *f_z_* were tested, by ranging the parameters between 0.5 µm/tooth and 7 µm/tooth with an increment of 0.5 µm/tooth between two consecutive tests. Feed per tooth is the parameter that determines the chip cross-section and consequentially its values have been chosen to investigate both the ploughing cutting regime and the shearing regime and the transition among them to extend the validation of the innovative procedure.

The micro-milling tests were performed on a five-axis Nano Precision Machining Centre (KERN Pyramid Nano, Kern Microtechnik GmbH, Eschenlohe, Germany) equipped with a Heidenhain iTCN 530 numeric control. The experimental tests were performed on the as-built surface achieved by the LB-PBF process. The load cell was a Kistler 9317C (Kistler Instrumente AG, Winterthur, Suisse), a piezoelectric 3-component force sensor, interfaced to three charge amplifiers (Kistler 5015A, Kistler Instrumente AG, Winterthur, Suisse). The force-measuring system accuracy is equal to 0.1 N and the sampling rate is 20 kHz. The natural frequency (2145 Hz for *F_x_* and 2192 Hz for *F_y_* as characterized in [35]) is lower than the tooth path frequency. The sample is fixed to the load cell with two screws. From the signal, a portion corresponding to thirty tool rotations was extrapolated to calculate the average cutting force signal. The optical measurement instrument used to determine micro-channels widths and depths was a Mitaka PF60 laser profilometer probe and in Table 5 its main properties are presented.

#### 2.2.3. Tool Run-Out Measurement

The micro-slot theoretical width is approximated to be twice the r_CE1_ value (Figure 4a). The laser-scanning speed used in the experimental measure was 20 µm/s and the measuring software was MountainsMap^®^ Premium version by Digital Surf. As Figure 4b and c show, each of the measures presented in the Results section is the result of the mean of different measures. Considering that the mean micro-slot length is equal to 4500 µm, every micro-slot width measured is the mean of five measures that have been made every 1125 µm (Figure 4b). At the same time, the depth of the channels has been estimated from the mean of 3 measurements taken every 2250 µm of the specimen length (Figure 4c). The mechanical deburring was performed on the specimen before the width measurement while the depth measures were made from the original specimen to avoid measure falsification.

Dealing with the cutting force signal, a Matlab script was elaborated to detect the difference between the two cutting force peaks in the y-direction. As highlighted in Figure 1, the phase angles of the two cutting edges are not constant and equal to *π*. Referring again to Figure 1, the *CE*1 phase angle is equal to the difference between the full angle (2*π*) and the phase angle *α* of *CE*2. This implicates different cutting times for *CE*1 and *CE*2, defined as *T*_1_ and *T*_2_, respectively. Considering the behavior of the theoretical cutting force component *F_y_* during the total cutting period *T* (the time which is spent by the tool to complete one full rotation), reported in Figure 5, *T*_1_ and *T*_2_ can be measured as the time difference between the two sequential valleys of the force signal. As a consequence, applying the equations presented in [21], the phase angles of *CE*1 and *CE*2 can be determined as fractions of the full angle. In particular, *α* is the full angle fraction related to the period fraction *T*_2_, as reported in Equation (36). Figure 5 also depicts the difference between the peaks of the two cutting edge forces Δ*F_y_*.

(36)
$$\alpha =2\pi \cdot \frac{T_{2}}{T}$$


## 3. Results and Discussion

In this section, the experimental results are presented together with the calculation of the run-out parameters by using both the theoretical equation and the experimental measurements. In the next subsection, the described analytical model is applied to the study case while a comparison between experimental values and analytical ones is proposed afterwards. In the last part of this section, a simplification of Equation (30) from a 4th grade equation to a 2nd grade model is derived from mathematical consideration and approximations. 

### 3.1. Experimental Measures

In Table 6, the width and depth measures of micro-slots are presented. The tool diameter was always measured before the execution of the tests and it was observed that the diameter was equal to 789 ± 2 μm. An ideal milling process generates a micro-slot width equal to twice the tool effective radius and a micro-slot width measure of 0.789 mm reflects a lack of run-out effect. Consequentially, the presence of run-out can be found where the width measure is higher than the ideal value just introduced. However, none of the measures has been discarded because they are also affected by the presence of burrs; a micro-slot width lower than 0.787 mm (which is exactly the minimal value of the tool’s effective diameter) means that the laser profilometer measured the distance between two irregular surfaces covered by burrs. In Figure 6, there is one of the images of the micro-slots after deburring, obtained by the multifocal 3D optical microscope. 

From the average width values, it could be observed that the variation of the feed per tooth values used in the tests has a neglectable effect on the measures; the difference between *r_CE_*_1_ maximum and minimum values is lower than 0.02 mm. Furthermore, in the literature [24,26], the value of feed per tooth considered as the limit in the transition from the ploughing regime to shearing is between 1.5 µm/tooth and 2 µm/tooth. From these statements, it could be said that the conclusion of the method presented will be associated with both regimes. Analyzing the measures presented in the previous table, it could be seen that eight of the fifteen micro-slot average width measures are lower than 0.787 mm.

Following that, the Matlab script was applied to the cutting force data to estimate the differences between force peaks in the y-direction. Equation (36) was used to compute the α for each micro-machining test. The results are collected in Table A1, reported in Appendix A, while the graphical representation of phase angles with respect to the peak force differences with their standard deviations is depicted in Figure 7. From a theoretical point of view, the test more affected by tool run-out is the one with the higher Δ*F_y_* peak and r_0_ peak while the value of the *α* angle is the lowest (it means it will be more different from the ideal 180°). From Figure 7, it can be observed that the test with *f_Z_* equal to 3.0 µm (number 8) is the one where there are the maximum Δ*F_y_* and the minimum *α*. Moreover, the tendency of *α* to reach the ideal value of 180° while decreasing Δ*F_y_* is visible as well, as expected by theoretical considerations.

The run-out parameters have been calculated from Equations (1)–(4) by using the experimental value for *r_CE_*_1_, *d*, and *α*. In Table 7, there are the main tool run-out parameters (*r*_0_ and *γ*) calculated from the tests. 

### 3.2. Application of the Analytical Model

The implementation of the model (Equation (30)) needs the calibration of the *K_ts_* variable for the AlSi10Mg alloy before the calculation of *r_CE_*_1_ values. Therefore, to evaluate the influence of *K_ts_* value on the model constants in Equation (30), a sensitivity analysis of *K_ts_*, in a range from 1 to 10^6^ N/mm^2^, has been performed. The results are reported in Table A2 where the first outcome is that the terms which assume the highest value are *g* and *c∙*(*r_CE_*_1_)^2^. In the literature, common values of *K_ts_* for metallic alloys are higher than 10^3^ N/mm^2^ and it can be supposed that AlSi10Mg has analogue *K_ts_*. Assuming this hypothesis (*K_ts_* > 10^3^ N/mm^2^), the order of magnitude of *g* and *c*∙(*r_CE_*_1_)^2^ is 10^−2^, while the other terms range between 10^−5^ and 10^−8^. The analysis also shows how higher values of *K_ts_* lead to lower values for the Equation (30) terms, as the influence of *K_ts_* on the model variables is reduced. In conclusion, the terms *a∙*(*r_CE_*_1_)^4^, *b∙*(*r_CE_*_1_)^3^, and *e*∙(*r_CE_*_1_) are approximated to zero. 

In Figure 8, the non-neglectable terms of Equation (30) are plotted as a function of *K_ts_*. In further calculations, a value of 10,000 N/mm^2^ is used for *K_ts_* as it is clear from Figure 8 that *K_ts_* values higher than 10 N/mm^2^ do not affect Equation (30).

### 3.3. Comparison between the Results

For estimating the analytical value of *r_CE_*_1_, the parameters of Equation (30) have been calculated for each experimental test. Equation (29) has been applied for the calculation of *I* with a value o*f K_ts_* equal to 10,000 N/mm^2^. For completeness, all the terms are reported in Table A3. The parameters *a*, *b*, *c*, *e*, and *g* have been determined from *I* and used together with experimentally measured *d*, *α*, Δ*F_y_*, *a_p_* parameters to solve the 4th grade equation. The solving algorithm applied to find the analytical values for *r_CE_*_1_ is the Lin–Bairstow method; the results are presented in Table 8, where there is also a comparison between *r_CE_*_1_ experimental values and the ones analytically computed. For each test, the Δ *variation* is the difference between the analytical *r_CE_*_1_ values and the experimental ones while the percentage error is the Δ *variation* divided by the analytical *r_CE_*_1_.

As shown in Table 8, and considering the first three decimals, the value of *r*_*CE*1_ calculated in all the experimental tests is the same and it is close to half the theoretical tool diameter (0.3945 mm).

### 3.4. Simplification of the Model

The sensitivity analysis demonstrates that only two parts of Equation (30) could not be approximated to zero. In Equation (37), Equation (30) is proposed without the null constants *a*, *b*, and *e*.

(37)
$$c\cdot {r_{CE1}}^{2}+g=0$$


This 2nd grade equation leads to a simplified equation to predict *r_CE_*_1_:
(38)
$$r_{CE1}=\pm \sqrt{-\frac{g}{c}}$$


From Table A2, it could be observed that the values for *g* and *c∙*(*r_CE_*_1_)^2^ are nearly the same between all tests; it demonstrates why the model results are not heavily affected by the variation of the feed per tooth. In Table 9, there is a comparison between the *r_CE_*_1_ values calculated from the complete 4th grade equation and the ones from the simplified 2nd grade model. As is shown, the difference between the two equations is always lower than 0.07% so the simplified version of the analytical method could be considered valid.

Moreover, Table 9 reveals that both the formulations of the analytical model (4th grade and 2nd grade equations) lead to the calculation of *r_CE_*_1_ values nearly identical to half the tool diameter, regardless of the cutting condition. This result does not agree with the reality, because that condition reflects the lack of tool run-out effect in every test while the experimental measures of *r_CE_*_1_ in Table 6 depend on the process parameters. For this reason, it has been proved that the analytical model elaborated is not effective in tool run-out calculation.

## 4. Conclusions

In this work, a methodology for evaluating the tool run-out based on the peak force difference between the first and second cutting edge in micro-milling is presented. Starting from a mechanistic model of micro-milling forces, already validated in previous research, 4th and simplified 2nd grade equations in the first cutting edge radius (*r_CE_*_1_) domain have been derived. The solution of these equations gives the value of *r_CE_*_1_, necessary for the calculation of tool run-out parameters. Being able to derive both *r_CE_*_1_ and the cutting edge phase angle directly from force measurements should give the possibility of instantaneously estimating the amount of tool run-out. This could allow future online run-out monitoring and compensation.

However, the solutions of the proposed equations are not in accordance with the experimental *r*_*CE*1_ results, giving values close to micro-mill radius that differ by only tens of microns. The consequence of this finding is that an experimental procedure for the measure of micro-slots is still required in the study of the tool run-out effect in milling. It has also been proved that the experimental measure is badly affected by the presence of burrs; in fact, more than half of the experimental width measures were lower than the tool diameter. According to this, further developments of the analytical method for the prediction of run-out become more necessary. 

The reasons for the model incorrectness could be found in the use of a well-performing cutting tool and micro-milling machine. In good working conditions, the tool run-out effect will be weak so it becomes more difficult its study and measure it. Even if accurate instruments have been used, their resolutions could be a problem: the laser profilometer resolution in the plane (0.1 μm) is exactly equal to the first changing decimal in the analytical *r_CE_*_1_ values as the force-measuring system accuracy is equal to 0.01 N that is the first changing decimal in Δ*F_y_* values calculated. 

The study of more critical cutting conditions could lead to better results. Further studies could test cutting tools affected by severe wear and fixed to the spindle with a less accurate tool holder. In these conditions, the run-out effect is prominent so that it will be easily detectable in experiments. This case of study will require a completely different analytical model because the one presented in the research is based on the strong hypothesis of equal ploughing areas between the two cutting edges. This statement is justified when the cutting tool works in ideal conditions, but it is not applicable in harder cutting conditions.

In future research, the validity of further analytical models could be proved without any time-consuming experimental test but by the generation of a dataset for the Δ*F_y_* values. This will avoid the inclusion of the effect of the presence of burrs on the micro-slot width measures.

## Figures and Tables

**Figure 1 micromachines-15-00305-f001:**
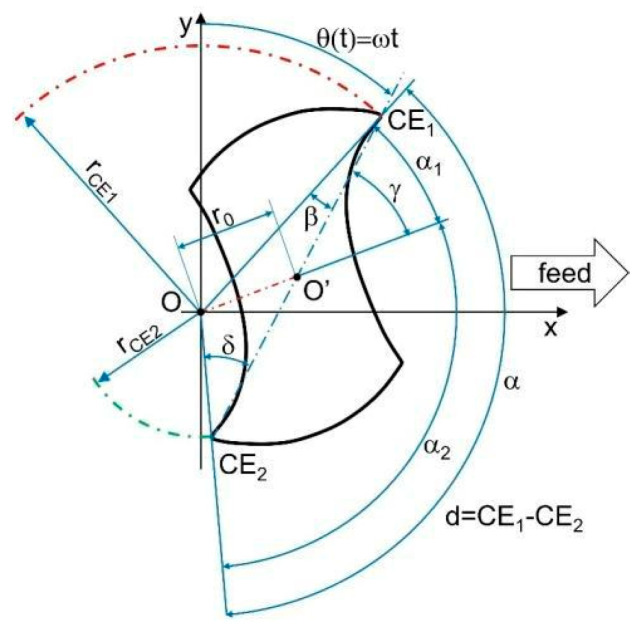
Geometrical representation of tool run-out.

**Figure 2 micromachines-15-00305-f002:**
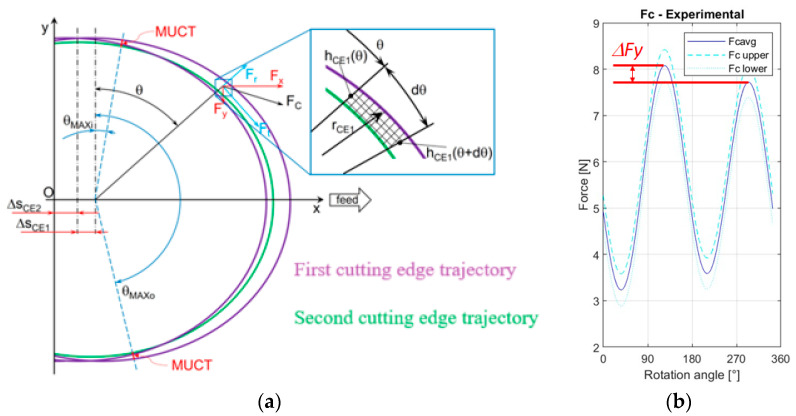
Cutting forces in micro-milling: (**a**) Cutting edge trajectories, instantaneous uncut chip thickness (IUCT) (*h_CEi_*(*θ*)), minimum uncut chip thickness (MUCT), and force components represented on the plane orthogonal to tool axial axis; (**b**) Trend of total cutting force (*F_c_*) vs. tool rotational angle (*θ*).

**Figure 3 micromachines-15-00305-f003:**
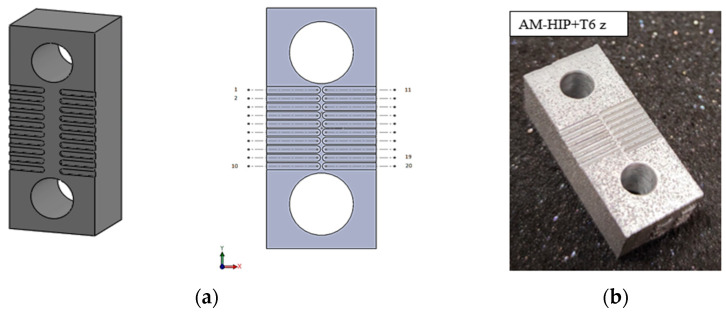
The AM specimen with the micro-channels made from micro-milling: (**a**) sample 3D drawing and the micro-slot scheme; (**b**) photo of the real specimen.

**Figure 4 micromachines-15-00305-f004:**
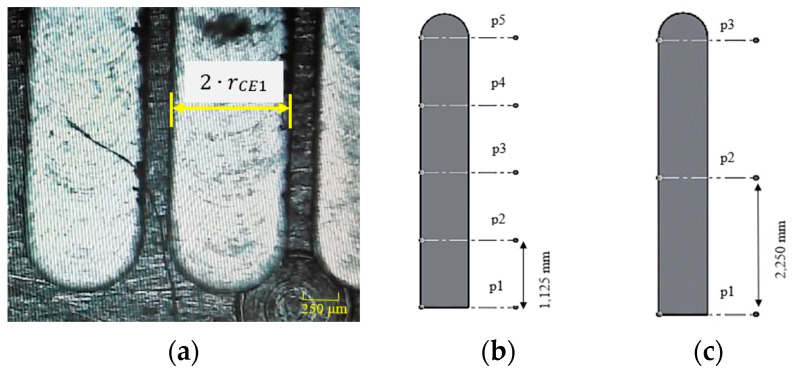
Experimental measures on the specimen: (**a**) value of r_CE1_ derived from the micro-slot width; (**b**) width measure acquisition scheme; (**c**) depth measure acquisition scheme.

**Figure 5 micromachines-15-00305-f005:**
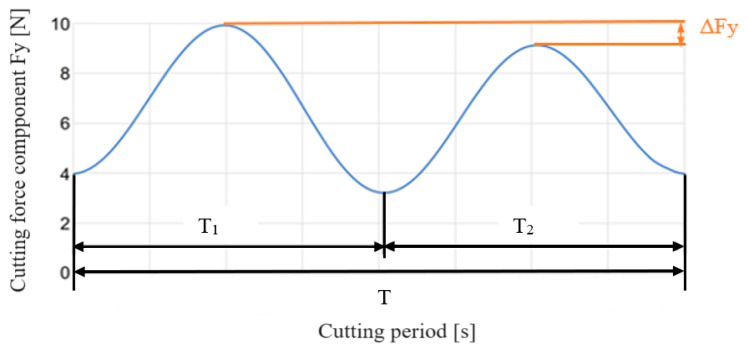
Theoretical cutting force signal as a function of time.

**Figure 6 micromachines-15-00305-f006:**
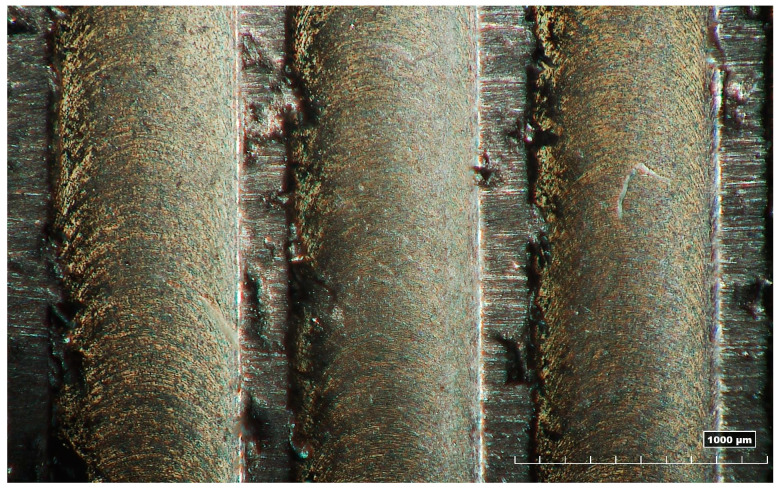
Micro-slot image obtained from the microscope after deburring.

**Figure 7 micromachines-15-00305-f007:**
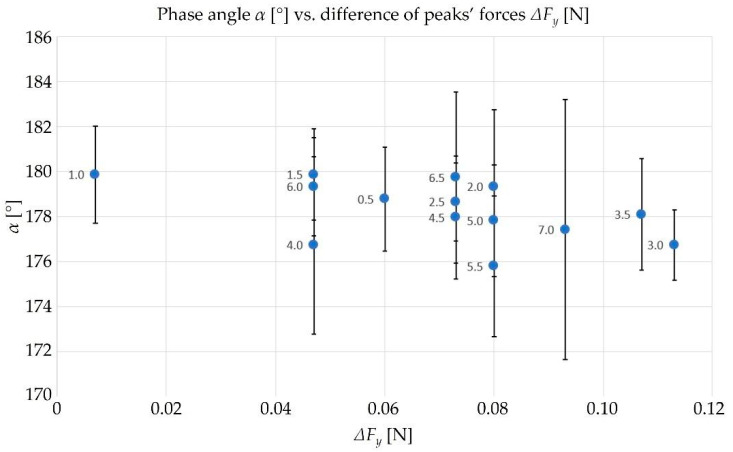
Experimental values of *α*, and related standard deviation, as a function of Δ*F_y_*. For each test, represented by a dot, the correspondent value of *f_Z_* is reported.

**Figure 8 micromachines-15-00305-f008:**
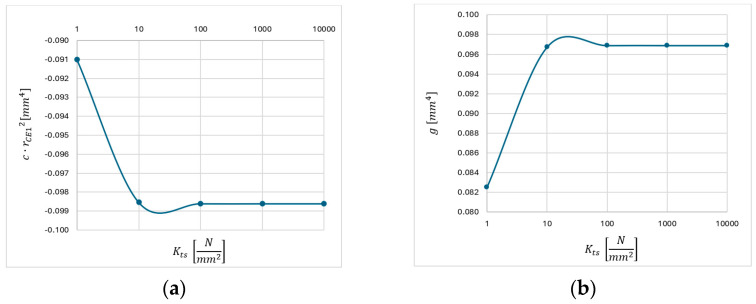
Trends of (**a**) variable *g*; (**b**) factor *c∙*(*r_CE_*_1_)^2^ as a function of K_ts_.

**Table 1 micromachines-15-00305-t001:** Chemical composition of AlSi10Mg (weight percentage).

AlSi10Mg	Si	Fe	Mn	Mg	Cu	Al
LB-PBF	10.2	0.21	<0.02	0.40	<0.002	Balance

**Table 2 micromachines-15-00305-t002:** LB-PBF process parameters.

Parameter	Value
Laser power [W]	370
Scanning speed [mm/s]	1300
Hatching distance [µm]	190
Layer thickness [µm]	30
Building platform temperature [°C]	80
Scanning direction	Vertical (z)

**Table 3 micromachines-15-00305-t003:** Properties of the tool employed for the micro-slot machining.

Property	Value
Model	103L008R005-MEGA-64-T
Nominal diameter [µm]	800
Effective diameter [µm]	789 ± 2
Nom. tool corner radius [µm]	5
Eff. tool corner radius [µm]	6.3
Helix angle [°]	20°
Rake angle [°]	4°
Material	Tungsten carbide
Coating material	Titanium nitride

**Table 4 micromachines-15-00305-t004:** Cutting parameters.

Parameter	Value
Cutting speed *v_c_* [m/min]	80
Axial depth *a_p_* [mm]	0.25
Feed per tooth *f_z_* [μm/tooth]	0.5–7.0

**Table 5 micromachines-15-00305-t005:** Laser profilometer properties.

Property	Value
Range measurement [mm]	60 × 60 × 10
x, y resolution [μm]	0.1
z resolution [μm]	0.01
Laser spot diameter [μm]	1
Laser wavelength [nm]	635

**Table 6 micromachines-15-00305-t006:** Measures of micro-slots machined on AM-HIP+T6 z specimen.

Test	f_z_ [mm/tooth]	f [mm/min]	Average Width [mm]	Dev St Width [mm]	Average a_p_ [mm]	Dev St a_p_ [mm]	Experimental r_CE1_ [mm]
1	0.0010	64	0.7604	0.0219	0.2607	0.0297	0.3802
2	0.0045	286	0.7755	0.0073	0.2373	0.0317	0.3878
3	0.0020	127	0.7816	0.0066	0.2385	0.0204	0.3908
4	0.0060	382	0.7816	0.0035	0.2502	0.0167	0.3908
5	0.0025	159	0.7662	0.0058	0.2391	0.0192	0.3831
6	0.0040	255	0.7792	0.0130	0.2374	0.0100	0.3896
7	0.0015	95	0.7730	0.0135	0.2545	0.0154	0.3865
8	0.0030	191	0.7960	0.0242	0.2590	0.0172	0.3980
9	0.0035	223	0.7896	0.0174	0.2616	0.0173	0.3948
10	0.0005	32	0.7879	0.0171	0.2641	0.0074	0.3940
11	0.0065	414	0.8021	0.0055	0.2676	0.0140	0.4010
12	0.0055	350	0.7829	0.0234	0.2541	0.0072	0.3915
13	0.0050	318	0.7951	0.0198	0.3119	0.0061	0.3975
14	0.0070	446	0.7997	0.0272	0.2477	0.0167	0.3998

**Table 7 micromachines-15-00305-t007:** Run-out parameters computed with the experimental *r_CE1_*.

Test	f_z_ [mm/tooth]	r_0_ [mm]	γ [rad]
1	0.0010	0.014	0.032
2	0.0045	0.010	0.803
3	0.0020	0.004	0.552
4	0.0060	0.004	0.557
5	0.0025	0.012	0.385
6	0.0040	0.012	1.149
7	0.0015	0.008	0.057
8	0.0030	0.012	1.281
9	0.0035	0.007	1.535
10	0.0005	0.004	1.440
11	0.0065	0.007	0.136
12	0.0055	0.015	1.346
13	0.0050	0.008	1.195
14	0.0070	0.010	1.038

**Table 8 micromachines-15-00305-t008:** Comparison between experimental and computed r_CE1_ values for each experimental test.

Test	fz [mm]	Experimental r_CE1_ [mm]	Analytical r_CE1_ [mm]	Δ Variation [mm]	Percentage Error [%]
1	0.0010	0.38018	0.39450	0.01432	3.63%
2	0.0045	0.38777	0.39460	0.00683	1.73%
3	0.0020	0.39082	0.39452	0.00370	0.94%
4	0.0060	0.39078	0.39452	0.00374	0.95%
5	0.0025	0.38308	0.39454	0.01146	2.91%
6	0.0040	0.38961	0.39470	0.00509	1.29%
7	0.0015	0.38651	0.39451	0.00799	2.03%
8	0.0030	0.39800	0.39470	−0.00330	−0.84%
9	0.0035	0.39479	0.39458	−0.00021	−0.05%
10	0.0005	0.39397	0.39453	0.00056	0.14%
11	0.0065	0.40104	0.39451	−0.00653	−1.65%
12	0.0055	0.39145	0.39484	0.00339	0.86%
13	0.0050	0.39754	0.39461	−0.00293	−0.74%
14	0.0070	0.39983	0.39466	−0.00517	−1.31%

**Table 9 micromachines-15-00305-t009:** Comparison between the analytical r_CE1_ values calculated from the 4th grade and the 2nd grade equations, for each experimental test.

Test	fz [mm]	4th Grade r_CE1_ [mm]	2nd Grade r_CE1_ [mm]	Δ Variation %
1	0.0010	0.39450	0.39450	0.00007%
2	0.0045	0.39460	0.39453	0.01591%
3	0.0020	0.39452	0.39451	0.00166%
4	0.0060	0.39452	0.39452	0.00173%
5	0.0025	0.39454	0.39452	0.00694%
6	0.0040	0.39470	0.39454	0.04082%
7	0.0015	0.39451	0.39451	0.00007%
8	0.0030	0.39470	0.39454	0.04046%
9	0.0035	0.39458	0.39453	0.01384%
10	0.0005	0.39453	0.39451	0.00568%
11	0.0065	0.39451	0.39451	0.00026%
12	0.0055	0.39484	0.39457	0.06759%
13	0.0050	0.39461	0.39454	0.01814%
14	0.0070	0.39466	0.39456	0.02531%

## Data Availability

The data that support the findings of this study are not openly available due to reasons of sensitivity and are available from the corresponding author upon reasonable request. Data are located in controlled access data storage at the University of Brescia.

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
