# Peer review of "Tool Run-Out in Micro-Milling: Development of an Analytical Model Based on Cutting Force Signal Analysis"

_micromachines, 2024, doi:10.3390/mi15030305_

Round 1
Reviewer 1 Report
Comments and Suggestions for Authors
This manuscript proposed a new method to calibrate the run-out model during micro-milling based on the cutting force signal elaboration. This paper will be attractive to researchers and engineers for micro-milling. The following questions should be addressed before future publication.
1. Introduction: The writing of the current research review is fine. The innovation of the paper's work has been distinguished from existing work. The reviewer suggests further refining the weaknesses of similar studies to better illustrate the necessity and urgency of this study.
2. Before the realization of the micro-slots, a single milling pass of rough machining is per-formed on the specimens to obtain a plane surface. Will the feed marks on the rough machining surface affect the cutting force of micro milling? If the impact is significant, subsequent grinding may be required to achieve smaller surface roughness. Although the process parameters for cutting are provided, it is recommended to provide the roughness value of the surface before micro milling.
3. The setup of experimental verification has not been provided. During the cutting force measurement, the sampling rate of 20 kHz is selected. Please explain the reason for choosing this high frequency. Usually, 2 kHz is sufficient for cutting force acquisition. Improve the description of force measurement. Some recent researches on force measurement during cutting are reported, which can be referred to support your method (see below). https://doi.org/10.1016/j.ultras.2023.107097. https://doi.org/10.1016/j.jmatprotec.2024.118296.
4. What is T1 in Figure 5?Besides, the difference of Fy between T1 and T2 should be mentioned. The description for Figure 5 is not enough, please improve.
5. Figure 6 shows the micro-slots image obtained from the laser profilometer with burrs. This image should have been taken by an optical microscope, not a laser profilometer. The laser profilometer is used to measure the size of the groove, and burrs basically hinder the measurement. Can you provide photos of micro-slots after deburring?
6. Improve the writing of conclusion. The discussion of Table 12 and some related results shoud be moved to Section 3.4. Simplification of the model. Besides, the novel findings can be further refined in conclusion.
Author Response
This manuscript proposed a new method to calibrate the run-out model during micro-milling based on the cutting force signal elaboration. This paper will be attractive to researchers and engineers for micro-milling. The following questions should be addressed before future publication.
Question 1. Introduction: The writing of the current research review is fine. The innovation of the paper's work has been distinguished from existing work. The reviewer suggests further refining the weaknesses of similar studies to better illustrate the necessity and urgency of this study.
Answer: The last part of the introduction section has been modified, in accordance with the reviewer suggestion, by underlining the limits of the available methodologies that, actually, can be exploited only by direct measurements and avoiding the possibility of an online monitoring strategy. Even if the online implementation of the methodology proposed in this work has not been performed yet, the authors highlighted this possibility as a future development.
Question 2. Before the realization of the micro-slots, a single milling pass of rough machining is per-formed on the specimens to obtain a plane surface. Will the feed marks on the rough machining surface affect the cutting force of micro milling? If the impact is significant, subsequent grinding may be required to achieve smaller surface roughness. Although the process parameters for cutting are provided, it is recommended to provide the roughness value of the surface before micro milling.
Answer: The statement indicated by the Reviewer was added to the text due to an error. The micro milling test were performed on the as built surface achieved by the AM process. The statement was corrected and highlighted in the text. The surface finishing of the as built sample was characterized and the Ra and Rz values were added to the text.
Question 3. The setup of experimental verification has not been provided. During the cutting force measurement, the sampling rate of 20 kHz is selected. Please explain the reason for choosing this high frequency. Usually, 2 kHz is sufficient for cutting force acquisition. Improve the description of force measurement. Some recent researches on force measurement during cutting are reported, which can be referred to support your method (see below). https://doi.org/10.1016/j.ultras.2023.107097. https://doi.org/10.1016/j.jmatprotec.2024.118296.
Answer: Thank you for the linked papers, we have checked them and added to the references list. About the sampling rate, in micro machining the tool path frequency is high and consequently the frequency of the signal could easily be in the order of some thousands of Hz. As consequence, we utilized a high sampling rate to avoid aliasing effect during the measurement. Further details were added to the description of the experimental apparatus to measure cutting force, and the reference [35] is an investigation dedicated to our cutting force measurement chain. All missing details can be found in the reference without excessively overload this publication.
Question 4. What is T1 in Figure 5?Besides, the difference of Fy between T1 and T2 should be mentioned. The description for Figure 5 is not enough, please improve.
Answer: In Figure 5, T1 is the cutting time of the first cutting edge CE1. The text has been modified by defining all the terms, namely T, T1, and T2, accordingly to the reviewer comment. In addition, a more detailed description of how the phase angle alpha has been determined, starting from the temporal Fy force signal, has been proposed. The meaning of the difference of Fy between T1 and T2 has been added as well. In general, the whole paragraph has been rewritten.
Question 5. Figure 6 shows the micro-slots image obtained from the laser profilometer with burrs. This image should have been taken by an optical microscope, not a laser profilometer. The laser profilometer is used to measure the size of the groove, and burrs basically hinder the measurement. Can you provide photos of micro-slots after deburring?
Answer: Figure 6 was edited as suggested by the Reviewer. A clear picture of the channels after deburring was acquired by using a 3D multifocal optical microscope.
Question 6. Improve the writing of conclusion. The discussion of Table 12 and some related results shoud be moved to Section 3.4. Simplification of the model. Besides, the novel findings can be further refined in conclusion.
Answer: The discussion of Table 12 results has been moved in the Section 3.4 Simplification of the model, as suggested. The conclusions have been slightly improved by underlining the possibility of a further implementation of the model for an online monitoring and compensation of tool run-out.

Reviewer 2 Report
Comments and Suggestions for Authors
In the domain of micro-machining, a pivotal finishing process for the precision fabrication of biomedical devices, the transition from conventional to micro-scale machining presents a number of challenges. Among these challenges, the accurate investigation of tool run-out stands out as particularly intricate, impeding the optimization of the micro-machining process.
The manuscript embarks on a focused analytical exploration of tool run-out, introducing an innovative method for calibrating model parameters grounded in the elaborate analysis of cutting force signals. Addressing the current reliance on time-consuming geometrical measurements for run-out prediction, this work endeavors to liberate the analytical process from dimensional constraints.
The calibration procedure is rigorously examined using data derived from the micromachining of additively manufactured Al-Si-10Mg specimens. Anchored in robust mathematical foundations, this novel approach demonstrates promise as it seeks to redefine tool run-out analysis and holds the potential for further refinement and application in subsequent studies.
Overall, the manuscript is well-written and can be published in the Micromachines. There is an issue to be corrected before publication:
The section 3 is loaded heavily with tables. It is hard to see the logic behind the 15 experiments. Authors might better move some tables to appendix highlighting the main trends in the main text.
Comments on the Quality of English LanguageThere are few minor misprints:
"alloys exhibits" - page 2 line 60
"un-correct" to be replaced with incorrect? - page 2 line 81
consider inserting a verb: "trajectories [being] very similar" - page 6 line 216
Author Response
In the domain of micro-machining, a pivotal finishing process for the precision fabrication of biomedical devices, the transition from conventional to micro-scale machining presents a number of challenges. Among these challenges, the accurate investigation of tool run-out stands out as particularly intricate, impeding the optimization of the micro-machining process.
The manuscript embarks on a focused analytical exploration of tool run-out, introducing an innovative method for calibrating model parameters grounded in the elaborate analysis of cutting force signals. Addressing the current reliance on time-consuming geometrical measurements for run-out prediction, this work endeavors to liberate the analytical process from dimensional constraints.
The calibration procedure is rigorously examined using data derived from the micromachining of additively manufactured Al-Si-10Mg specimens. Anchored in robust mathematical foundations, this novel approach demonstrates promise as it seeks to redefine tool run-out analysis and holds the potential for further refinement and application in subsequent studies.
Overall, the manuscript is well-written and can be published in the Micromachines. There is an issue to be corrected before publication:
Question 1. The section 3 is loaded heavily with tables. It is hard to see the logic behind the 15 experiments. Authors might better move some tables to appendix highlighting the main trends in the main text.
Answer: As suggested by the reviewer, tables previously numbered as Table 7, Table 9, and Table 10 have been moved in the Appendix A. As a consequence, the new table numbering is:
- OLD Table 7 → NEW Table A1
- OLD Table 9 → NEW Table A2
- OLD Table 10 → NEW Table A3
- OLD Table 8 → NEW Table 7
- OLD Table 11 → NEW Table 8
- OLD Table 12 → NEW Table 9.
Question 2. There are few minor misprints:
"alloys exhibits" - page 2 line 60
"un-correct" to be replaced with incorrect? - page 2 line 81
consider inserting a verb: "trajectories [being] very similar" - page 6 line 216
Answer: The authors thank the reviewer for the observation. The misprints have been corrected as indicated. In addition, a general revision of the text has been performed.
